# Automatic Task-Level Thinking Steps Help Large Language Models for Challenging Classification Task

**Chunhui Du, Jidong Tian, Haoran Liao, Jindou Chen, Hao He, Yaohui Jin**

MoE Key Lab of Artificial Intelligence, AI Institute, Shanghai Jiao Tong University

{chunhui18, frank92, liaohaoran, goldenbean, hehao, jinyh}@sjtu.edu.cn

## Abstract

Large language models (LLMs) have shown incredible performance on many tasks such as dialogue generation, commonsense reasoning and question answering. In-context learning (ICL) is an important paradigm for adapting LLMs to the downstream tasks by prompting few demonstrations. However, the distribution of demonstrations can severely affect the performance, especially for challenging classification tasks. In this paper, we propose the concept of task-level thinking steps that can eliminate bias introduced by demonstrations. Further, to help LLMs distinguish confusing classes, we design a progressive revision framework, which can improve the thinking steps by correcting hard demonstrations. Experimental results prove the superiority of our proposed method, achieving best performance on three kinds of challenging classification tasks in the zero-shot and few-shot settings. Besides, with task-level thinking steps, automatically generated chain-of-thoughts (CoTs) bring more competitive performance.

## 1 Introduction

Large language models (LLMs) have shown incredible performance on many tasks such as dialogue generation, commonsense reasoning, and question answering (Dong et al., 2022). With the new paradigm of in-context learning (ICL) (Brown et al., 2020; Pan et al., 2023), LLMs directly adapt to downstream tasks without updating parameters by prompting with few demonstrations. Wei et al. (2022) found that elaborating the reasoning steps in demonstrations can significantly stimulate the complex reasoning ability of LLMs, which is called manual chain-of-thought (CoT). Zero-shot-CoT (Kojima et al., 2022) enables LLMs to generate thinking steps automatically using the simple but efficient prompt "*Let's think step by step.*", as shown in Figure 1(a). More recently, plan-and-solve prompt (Wang et al., 2023) extends it with

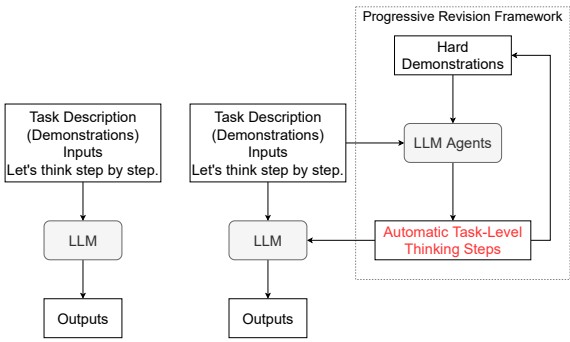

Figure 1: Zero-shot-CoT Prompt (Kojima et al., 2022) vs. Prompt with Automatic Task-Level Thinking Steps

"*Let's first understand the problem, devise a plan, and solve the problem step by step.*" to address the issue of missing reasoning steps and "*extract relevant variables and their corresponding numerals*" to address the issue of calculation errors. Inspired by this, we propose the concept of **task-level thinking steps** for challenging classification tasks as shown in Figure 1(b), which helps LLMs to clarify tasks and remind them of overlooked issues. To this end, we design a progressive revision framework, which automatically generates thinking steps by LLM agents and then progressively revise the steps by correcting hard demonstrations.

For classification tasks, it is widely investigated that the demonstrations distribution, including content, label, and even the order, can severely affect the performance (Lu et al., 2022a; Turpin et al., 2023; Zhao et al., 2021). Therefore, many works focused on selecting high-quality and representative demonstrations, such as selecting similar demonstrations (Liu et al., 2021) or maintaining the class balance between demonstrations (Lu et al., 2022b). However, the best distribution depends on the task, and bias is inevitable. Our proposed task-level thinking steps are helpful in reducing the bias introduced by demonstrations. Table 1 shows an example of the stance detection task. Zero-shot-CoT

| |
|---|
| **Task description**: Perform stance detection. classes: (A)none (B)against (c)favor |
| **Input**: For domain abortion, You can't give the "death penalty" to something not living sentient or autonomous. |
| **No Demonstrations** 
 **Output of zero-shot-CoT prompt** 
 The stance in this context is (C) favor, as the statement suggests that the speaker is in favor of abortion rights. The argument is that a fetus is not a living, sentient, or autonomous being and cannot be subject to the death penalty. ✓ |
| **Biased Demonstrations**: input:[...] answer:(B) input:[...] answer:(B) input:[...] answer:(B) 
 **Output of zero-shot-CoT prompt** 
 First, the statement is discussing abortion. Next, the statement is arguing against the idea of giving the "death penalty" to something that is not living, sentient, or autonomous. This suggests a stance against abortion. ✗ 
 **Output with task-Level thinking steps** 
 1. Identify the domain abortion discussed in the statement. 2. Look for keywords or phrases ... the speaker is against the idea of considering a fetus as a living, sentient, or autonomous being. 3. Determine the stance of the statement. The statement is against the idea of considering fetus as a living, sentient, or autonomous being, which suggests that the speaker is in favor of abortion. 4. Provide the answer to the stance detection task which is (C) in favor of abortion. ✓ |

Table 1: Zero-shot-CoT prompt is affected by the biased demonstrations while prompt with task-level thinking steps outputs correctly. Omitted demonstrations are shown in Appendix A.

predicts correctly with no demonstrations. However, the output is severely affected by "*against*" from the biased demonstrations. The automatically generated task-level thinking steps include: identifying the domain; finding the viewpoint; determining the stance with respect to the domain; and matching the choices. With the help of these steps, LLMs perceive the implicit meaning "*a fetus is not a living, sentient, or autonomous being*" and make the correct prediction "*in favor of abortion*".

In addition to debiasing, we would like task-level thinking steps to clarify some confusing classes. To this end, we design a **progressive revision framework** to generate the thinking steps by correcting

hard demonstrations. In the framework, as shown in Figure 2, we set up two LLM agents (Andreas, 2022) with character settings and historical memories, the **teacher** agent and the **student** agent. For each hard demonstration, the student agent generates outputs based on the task-level thinking steps generated by the teacher agent. If the prediction is correct, move on to the next demonstration. If the prediction is wrong, the teacher agent first analyses the error cause and then tries to revise the task-level thinking steps. For the example in Figure 2, class (B) and class (D) tend to be confusing. After the teacher agent's analysis, an explicit reminder can be inserted in a suitable place in the revised task-level thinking steps. Then, the student agent generates outputs again. The iteration may be repeated several times until the correct prediction or the maximum number of times allowed is reached.

In this paper, we consider three classification tasks: **multifaceted analysis of subjective text** requires first locating aspects, then identifying opinions, and finally analyzing polarity; **fine-grained text classification** requires a deep distinction between numerous and similar classes; **domain-specific classification** requires the understanding of domain-related questions, choices, and demonstrations. We conduct experiments on zero-shot, few-shot, and few-shot-CoT settings, specifically. Experimental results demonstrate the effectiveness of our proposed method by outperforming the competitive baseline models. Further ablation analyses and robustness experiments reveal the benefits and effects of each module.

Our contributions are threefold: 1) We propose the novel concept of task-level thinking steps, which can eliminate bias introduced by demonstrations 2) We design a progressive revision framework, which purposefully improves the thinking steps based on feedback from hard demonstrations for LLMs. 3) Our proposed method surpasses competitive baselines in the zero-shot, few-shot, and few-shot-CoT settings.

## 2 Methods

### 2.1 Task Definition

We consider a general classification task, where $T$ denotes the task description, $x \in \mathcal{X}$ is the text input, and $y \in \mathcal{Y}$ is the corresponding label. $|\mathcal{Y}| = C$ is the number of classes. $\mathcal{D}_{train} = \{(x_1, y_1), ..., (x_M, y_M)\}$ denotes the train dataset and $\mathcal{D}_{test} = \{(x_1, y_1), ..., (x_N, y_N)\}$ denotes the

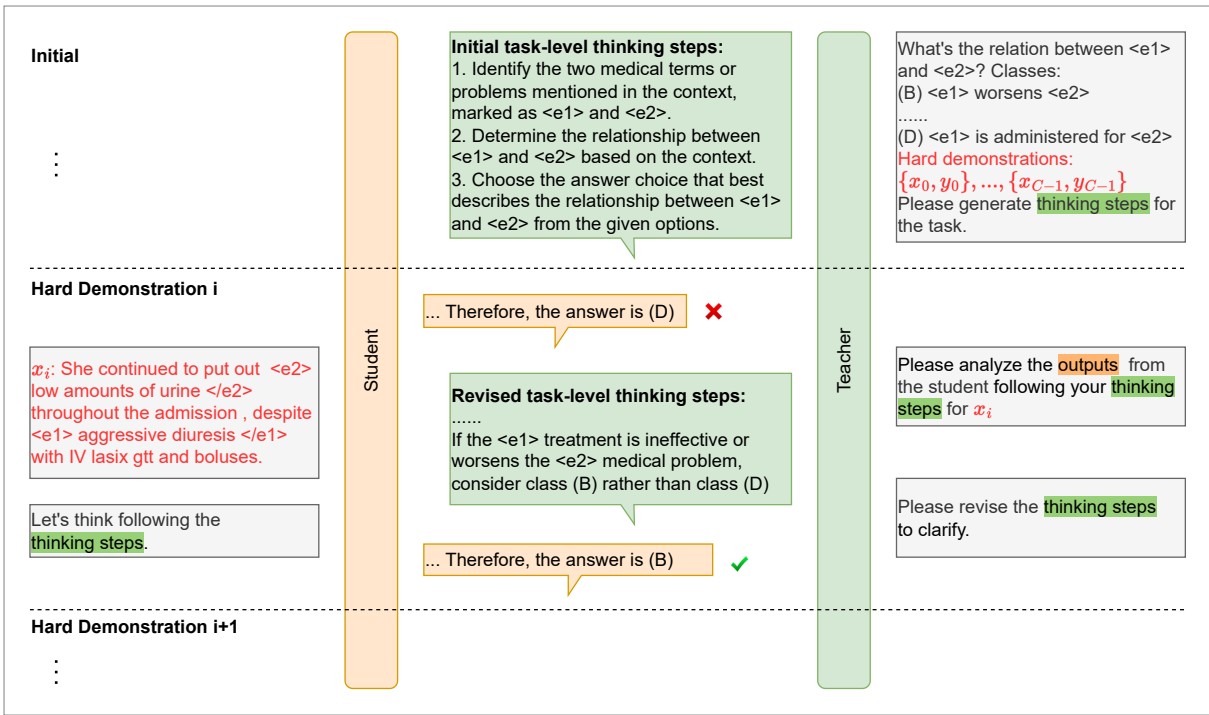

Figure 2: Progressive revision framework. There are two LLM agents: teacher and student. Rectangles represent the prompts to agents, outputs of the teacher agent and outputs of the student agent.

test dataset. An LLM with parameters $\theta$ performs zero-shot learning by conditioning on the task description $T$ and the test input $x \sim \mathcal{D}_{test}$. The likelihood of a candidate answer $y$ could be represented by a scoring function $f$:

$$P(y|x) = f_\theta(T, x) \quad (1)$$

When sampling demonstrations $\mathcal{D}_{demo}$ from $\mathcal{D}_{train}$, the likelihood for few-shot learning is:

$$P(y|x) = f_\theta(T, \mathcal{D}_{demo}, x) \quad (2)$$

The final predicted label $\hat{y}$ is the candidate answer with the highest probability:

$$\hat{y} = \arg\max_{y \in \mathcal{Y}} P(y|x_{text}) \quad (3)$$

Only hard demonstrations can generate valuable task-level thinking steps. In this paper, we traverse $\mathcal{D}_{train}$ and select one that the LLMs can not predict correctly for each class to construct $\mathcal{D}_{demo} = \{(x_0, y_0), ..., (x_{C-1}, y_{C-1})\}$ as shown in Figure 2.

## 2.2 LLM Agents

LLMs can serve as models of agents as suggested in (Andreas, 2022), which output responses based on initial setting, historical memories, and current observations. We set up two LLM agents. The **teacher** agent is responsible for generating the task-level thinking steps from hard demonstrations with labels. In addition, the teacher agent needs to analyze the errors in the student agent's answers and revise the task-level thinking steps. The **student** agent is responsible for giving answers and explanations for each presentation with the task-level thinking steps provided by the teacher agent.

## 2.3 Progressive Revision Framework

Given hard demonstrations $\mathcal{D}_{demo}$, the teacher agent first generates initial thinking steps $S_0$, then progressively revise it based on the outputs from the student agent as shown in Figure 2.

**Generate initial task-level thinking steps.** The teacher agent uses hard demonstrations to generate initial task-level thinking steps. The prompt is "*Please generate generic thinking steps for the task.*". Taking the medical relation extraction task as an example, here are the generated initial task-level thinking steps " *1. Identify the two medical terms or problems mentioned in the context, marked as <e1> and <e2>. 2. Determine the relationship between <e1> and <e2> based on the context. 3. Choose the answer choice that best describes the relationship between <e1> and <e2> from the given options.* ".

We can see that the teacher agent understands the task description and gives three reasonable steps: identifying two entities, determining the relation, and matching the classes. However, they are also preliminary and abbreviated.

**Generate outputs.** For the $i$-th hard demonstration $x_i$, the student agent receives the task-level thinking steps $S_i$ and generates outputs $R_i$. The prompt is "*Thinking steps: {$S_i$}. Let's think following the thinking steps.*" As shown in Figure 2, the correct answer is "*(B) <e1> worsens <e2>*" but the student agent outputs "*The answer is (D)*" and sends it to the teacher agent.

**Improve task-Level thinking steps.** The teacher agent analyzes the reason for the error with the prompt "*Input: {$x_i$}. Outputs: {$R_i$}. Please analyze the outputs from the student following your thinking steps.*". Then the teacher agent revises the task-level thinking steps with the prompt "*Please revise the thinking steps to clarify.*".

We find that the revised thinking steps do not usually change much but rather extend more reminders. The main extension in this example is " *If the treatment is ineffective or worsens the medical problem, consider choice (B) rather than choice (D).*". It can be seen that the student agent only focuses on the causal relationship between the treatment and the medical problem while does not pay attention to the effects. In fact, this input places more emphasis on the effect of the treatment on the disease than on how the treatment should be taken for the disease.

**Generate outputs with revised thinking steps.** With the revised thinking steps $S_i$, the student agent predicts correctly. Then, the iteration proceeds to the next hard demonstration. Otherwise, the output is sent to the teacher again until the correct prediction or the maximum attempts are reached. Experiments show that even if the hard demonstration was not successfully corrected, the revised thinking step was still useful.

**Checking mechanism.** We find that LLMs, especially *gpt-4*, are effective in correcting hard samples by revising task-level thinking steps. However, this may be achieved by extending shortcuts rather than trustworthy analysis. For example, "*If you are not sure, choose (B).*". Therefore, we set up a checking mechanism. The teacher agent tries to generate multiple candidate thinking steps that can successfully correct $x_i$. The student agent tests performance on $\{x_0, ..., x_{i-1}\}$ and keeps the best one.

## 2.4 Automatic Chain-of-Thought

Another advantage of our proposed task-level thinking steps is that is very helpful to generate reliable CoTs for demonstrations automatically. Manual CoTs are labor intensive, and the human annotations do not necessarily match LLMs. Zero-shot-CoT, while avoiding human engineering, may produce unfaithful explanations especially for hard demonstrations.

## 3 Experiments

### 3.1 Datsets

We choose three kinds of challenging classification tasks. The statistics and descriptions of all tasks are shown in Table 7 and Table 8 in Appendix B.

**Multifaceted analysis of subjective text** is the task that involves different aspects of subjective human feeling reflected in the text. The challenge is recognizing and understanding a broader range of human emotional states rather than identifying common positive or negative feelings. Following Zhang et al. (2023a), we use Stance16 (Mohammad et al., 2016) for stance detection and Emotion20 (Barbieri et al., 2020) for emotion recognition.

**Fine-grained text classification** is the task of categorizing text or documents into predefined topics or categories. We carefully select 20news [1] dataset. The challenge lies in numerous and similar classes.

**Domain-specific classification** adopts the medical relation classification task, which refers to identifying the relationship between pairs of entities within medical texts. The challenge is the need for domain knowledge, and differences between relations are subtle. i2b2 (Uzuner et al., 2011) collects clinical records and is used to classify relations between medical problems, medical tests, and treatments. ChemProt (Krallinger et al., 2017) is used to determine how chemical acts on the protein, such as activation and inhibition. DDI (Herrero-Zazo et al., 2013) is used to determine the interaction relation between two drugs, such as pharmacokinetics and pharmacodynamics.

### 3.2 Experimental Settings

We constructed experiments to evaluate the effectiveness of thinking steps. $C$ hard demonstrations are sampled for each task to generate thinking steps. The LLM for the teacher agent is *gpt-4*, the LLM for the student agent and the LLM for running

---

[1] http://qwone.com/~jason/20Newsgroups/

| Methods | | Stance16 | Emotion20 | 20news | i2b2 | ChemProt | DDI | Avg. |
|---|---|---|---|---|---|---|---|---|
| Zero-shot | Default | 41.49 | 70.02 | 45.99 | 14.89 | 41.97 | 49.17 | 43.92 |
| | Auto | 47.62 | 72.81 | 39.97 | 29.63 | 54.47 | 36.22 | 46.78 |
| | ITS | 41.66 | 68.96 | 63.30 | 43.82 | 50.91 | 39.57 | 51.37 |
| | PRTS | **52.38** | **75.76** | **67.20** | **54.99** | **68.33** | **57.36** | **62.67** |
| Few-shot | Default | 50.04 | 71.58 | 56.29 | 35.54 | 43.03 | 46.13 | 50.43 |
| | Auto | 51.09 | 75.28 | 41.76 | 42.62 | 65.73 | 49.29 | 54.30 |
| | ITS | 36.06 | 74.01 | 53.56 | 40.48 | 49.01 | 50.15 | 50.54 |
| | PRTS | **56.35** | **77.82** | **66.58** | **52.30** | **69.67** | **62.45** | **64.19** |
| Few-shot-CoT | Auto | 37.17 | 72.72 | 42.02 | 40.94 | 67.40 | 50.73 | 51.83 |
| | ITS | 40.39 | 75.21 | 58.71 | 60.50 | 76.72 | 63.23 | 62.46 |
| | PRTS | **60.30** | **78.92** | **72.35** | **67.45** | **79.20** | **76.67** | **72.47** |

Table 2: The overall performance of PRTS and baselines on six classification tasks. **Bold** denotes the best in corresponding settings specifically.

experiments are both *gpt-3.5-turbo*. We set a temperature of 0 for all experiments. For each hard demonstration, the maximum number of revision attempts is 5. For the checking mechanism, the number of candidates is 3. We report the F1-score as the evaluation metric.

The followings methods serve as baselines: **Default** generates outputs directly and **Auto** generates outputs with the prompt "*Let's think step by step*" (Kojima et al., 2022). Our methods generate outputs with Initial Thinking Steps (**ITS**) to validate the effectiveness of thinking steps and Progressively Revised Thinking Steps (**PRTS**) to validate the effectiveness of framework. For few-shot, prompt demonstrations are the same as hard demonstrations to avoid unfair comparisons. In Section 4.2, we construct experiments with other prompt demonstrations. For few-shot-CoT, Auto, ITS and PRTS can automatically explain the prompt demonstrations without human efforts following (Wang et al., 2023; Shum et al., 2023). Detailed prompt templates are shown in Table 9 in Appendix B.

## 4 Results

### 4.1 Main Results

The experimental results are shown in Table 2. All results are averaged over three runs. Overall, PRTS performs best on all tasks, which outperforms Auto by 18.75% for zero-shot, 9.89% for few-shot, and 20.64% for few-shot-CoT on average.

For the zero-shot setting, ITS has slightly better performance than Default and Auto on average, but which one is better depends on the task. Specifically, ITS outperforms Auto by 23.33% for 20news and by 14.19% for i2b2. One possible reason is that

both 20news and i2b2 have many similar classes, and the thinking steps include analysis and comparison between classes, which may help in the fine-grained classification of LLMs. With progressive revision, PRTS performs best on all tasks.

For the few-shot setting, PRTS outperforms Auto by an average of 9.89%. It is worth noting that ITS has worse performance than Auto. One possible reason is that initial thinking steps may be contradictory to ICL for LLMs. However, we will show that significant improvements can be obtained by explaining these demonstrations with thinking steps.

For few-shot-CoT, PRTS outperforms Auto by 20.64% and ITS by 10.01% on average. For Auto, Stance16 and 20news have worse performance compared with the few-shot setting. One possible reason is that LLMs may produce reluctant explanations and hallucinations for hard demonstrations. Table 4 shows the automatically generated CoT by Auto, ITS, and PRTS: For **Auto**, it can be seen that the LLM can not understand the stance detection task. Therefore, the LLM fails to locate the domain word "*atheism*" and mistakenly believes the answer is "*(C) favor*". However, faced with the golden label "*(B) against*", the LLM can only generate the reluctant and absurd explanation: "*against (B) is the closest to the opposite of favor*". With this wrong CoT demonstration, a worse performance than zero-shot and few-shot would make sense. For **ITS**, LLM successfully locates the domain word "*atheism*" and seems to explain the golden label correctly. However, LLM only states that "*against is the idea of freedom from religion*" but does not explicitly state that atheism is equivalent to free-

| Methods | Stance16 | 20news | i2b2 |
|---|---|---|---|
| Similar | 51.82 | 36.63 | 47.86 |
| Similar+PRTS | **53.97** | **59.20** | **52.91** |
| Unbalanced | 41.78 | 21.98 | 12.93 |
| Unbalanced+PRTS | **52.38** | **65.09** | **42.36** |
| Easy | **53.09** | 51.20 | 27.90 |
| Easy+PRTS | 51.79 | **67.33** | **52.98** |

Table 3: PRTS consistently improves performance for various sampling methods of prompt demonstrations.

dom from religion. This may lead LLMs to ignore domain words and only decide on the sentence. For **PRTS**, the CoT is more consistent and complete and always revolves around the domain word "*atheism*". COT explicitly explains that "*a stance against atheism*" is because "*people should not be free from religion*".

## 4.2 Ablation Studies

**Analysis of demonstrations distribution.** In this section, we explore the effects of various sampling methods of prompt demonstrations. For the few-shot setting, **Similar** (Liu et al., 2021) dynamically selects $C$ most similar for each sample. **Unbalanced** selects prompt demonstrations in the same class, which is considered as biased (Zhao et al., 2021). **Easy** is the same as our hard demonstration selection, which selects one that LLMs predict correctly for each class. Results are reported in Table 3 for Stance16, 20news, and i2b2. We can find that which sampling method is better depends on the task, but PRTS consistently improves by 9.93% for Similar, 27.71% for Unbalanced, and 13.3% for Easy, respectively. Especially compared with Unbalanced performing poorly on all tasks, Unbalanced+PRTS remains very robust. This shows that PRTS can eliminate bias introduced by demonstrations effectively.

**Analysis of the progressive revision framework.** Our key idea is revising the task-level thinking steps by correcting the hard demonstrations, thus enhancing the ability to distinguish confusing classes. Specifically, we first correct the hard demonstrations and then further improve thinking steps through the Checking mechanism.

While correcting the hard demonstrations of class $c$, it is also beneficial to all samples. We validate this idea on two datasets for zero-shot setting: Stance16 and i2b2 as shown in Figure

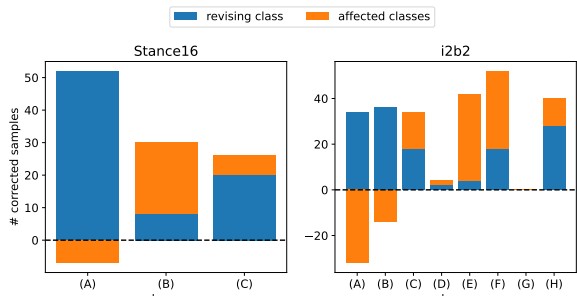

Figure 3: The number of corrected samples when correcting hard demonstrations of each class.

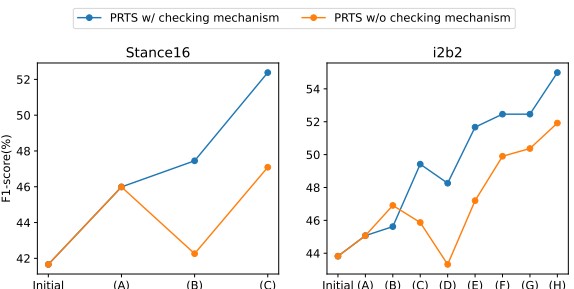

Figure 4: Performance changes compared PRTS w/ checking mechanism with PRTS w/o checking mechanism in progressive revising thinking steps.

3. The class $c$ of the hard demonstration being corrected is named **revising class** while the remaining classes $\{1, ..., c-1, c+1, ..., C\}$ are named **affected classes**. It is always helpful for revising class and may also have a positive effect on affected classes.

In addition, we further improve thinking steps through the checking mechanism when iterating, i.e., progressive revising thinking steps. As shown in Figure 4, for PRTS w/o checking mechanism, huge performance degradation occurs for class (B) in Stance16 and class (D) in i2b2. For comparison, PRTS w/ checking mechanism almost improves progressively in each round and outperforms the PRTS w/o checking mechanism 5.29% for Stance16 and 3.07% for i2b2.

## 4.3 Robustness Experiments

**Effects of class names.** For classification tasks, class names can seriously affect the performance of LLMs. For 20news and DDI, the **default classes** are ambitious, and we have manually annotated them as **manual classes**. In this section, we explore the effects of different class names. More details are shown in Table 8 in Appendix B.

We experimented on zero-shot, few-shot, and

| | input: For domain atheism, The framers of our Constitution meant we were to have freedom of religion, not freedom from religion. - Billy Graham quote SemST |
|---|---|
| | classes: (A) none (B) against (C) favor answer: (B) |
| Auto | The statement suggests a pro-religion stance, implying that the person making the statement is in favor of religion. Therefore, the stance detection task is "favor" (C). However, the prompt specifically asks for stance detection from the given choices, and "favor" is not one of the choices. Among the given choices, "against" (B) is the closest to the opposite of "favor" |
| ITS | 1. The context is about atheism and a quote by Billy Graham 2. ... 3... 4. The stance expressed is against the idea of freedom from religion. 5. Therefore, the appropriate answer is (B). |
| PRTS | 1. Read and understand the context: ... 2. Identify the domain: The domain is atheism. 3. Determine the stance: ...This implies a stance against atheism, as it suggests that people should not be free from religion. 4. Consider the broader implications... 5. Choose the choice (B) |

Table 4: Case study for automatically generated CoT. Auto CoT is reluctant and absurd. ITS CoT does not explicitly state the relation between stance "*atheism*" and "*freedom from religion*". PRTS CoT is complete and always revolves around the domain word "*atheism*".

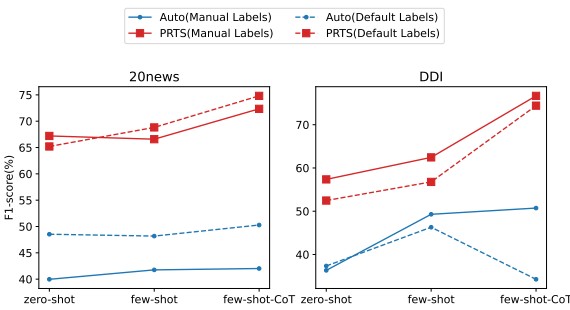

Figure 5: Performance on zero-shot, few-shot and few-shot-CoT settings for Auto and PRTS. The dashed lines indicate default classes and the solid lines indicate manual classes.

few-shot-CoT for Auto and PRTS. To our surprise, the default classes performed much better than the manual classes, especially for Auto on 20news, as shown in Figure 5(a). The reason may be that 20news is a pre-trained corpus for LLMs and is therefore more compatible with the default classes. Nevertheless, PRTS has a similar performance with manual classes and default classes.

For DDI, the performance on the default classes is particularly worse than on manual classes for Auto as shown in Figure 5(b). We find that LLMs can correctly recognize and interpret the medical terms such as "*pharmacokinetics*" and "*pharmacodynamics*" with the strong capability on medical problems (Nori et al., 2023). The poor performance is because default class "*int*" was mistakenly thought to be a useless and distracting choice. However, PRTS achieves similar performance to manual classes because the task-level thinking steps explicitly state that

"*int is an abbreviation for interaction except pharmacokinetics and pharmacodynamics*".

**Effects of LLMs.** We find that only *gpt-4* helps the teacher agent to revise thinking steps progressively. *text-davinci-003* is not a dialogue model and can not even produce reasonable initial thinking steps. *gpt-3.5-turbo* enables interaction between the student and the teacher, but it is difficult to generate valuable thinking steps to correct the hard demonstrations. For the hard demonstrators covered in this paper, *gpt-4* can almost revise thinking steps to correct them within five attempts.

Therefore, we keep the previous obtained thinking steps and rerun the experiment on *text-davinci-003* and *gpt-4* as shown in Table 5. Overall, the classification performance of *text-davinci-003*, *gpt-3.5-turbo*, and *gpt-4* improves sequentially for both Auto and PRTS. Interestingly, the performance improvements of the fine-grained classification task, 20news ($39.97\% \rightarrow 70.03\%$) and i2b2 ($29.63\% \rightarrow 58.33\%$), are huge from *gpt3.5-turbo* to *gpt-4* for Auto. It is not clear whether this is because *gpt-4* was pretrained on these datasets or its instruction tuning is similar to our thinking steps. Besides, all three LLMs perform poorly on Stance16 for Auto, while PRTS can get a huge boost on *gpt-4*. This suggests that PRTS can accurately analyze and disambiguate the task for ambitious tasks, then provide explicit thinking steps.

### 4.4 Cost Analysis

In this section, we analyze the additional cost of task-level thinking steps includes the **generation cost** based on hard demonstrations and **prompt**

| LLMs | | Stance16 | Emotion20 | 20news | i2b2 | ChemProt | DDI | Avg. |
|---|---|---|---|---|---|---|---|---|
| *text-davinci-003* | Auto | 44.44 | 65.29 | 32.45 | 12.81 | **40.69** | 25.20 | 36.81 |
| | PRTS | **45.35** | **70.91** | **57.55** | **54.64** | 39.94 | **41.96** | **51.72** |
| *gpt-3.5-turbo* | Auto | 47.91 | 75.28 | 39.97 | 29.63 | 54.47 | 36.22 | 47.25 |
| | PRTS | **52.38** | **77.82** | **67.20** | **54.99** | **68.33** | **57.36** | **63.01** |
| *gpt-4* | Auto | 41.81 | 78.64 | 70.03 | 58.33 | 60.71 | 48.01 | 59.58 |
| | PRTS | **66.76** | **80.20** | **78.33** | **66.59** | **68.33** | **76.54** | **72.79** |

Table 5: Zero-shot performance of PRTS and Auto for different LLMs.

| methods | generation tokens | prompt tokens |
|---|---|---|
| | **general dataset** | |
| Default | - | $T_{input} \times N_{test}$ |
| Auto | - | $(T_{input} + T_{task}) \times N_{test}$ |
| PRTS | $T_{input} \times H \times C + 2R \times (T_{task} + 2T_{input}) \times C$ | $(T_{input} + 2T_{task}) \times N_{test}$ |
| | **i2b2 dataset** | |
| Default | - | $T_{input} \times N_{test}$ |
| Auto | - | $3T_{input} \times N_{test}$ |
| PRTS | $168T_{input}$ | $5T_{input} \times N_{test}$ |

Table 6: Cost comparison for different methods.

**cost** for test inputs.

Generation cost includes the selection of hard demonstrations and iterative refinement. Assume that the average tokens of train/test inputs are $T_{input}$ (outputs with simple answers can be ignored). It takes an average traversal of $H$ train inputs to select hard demonstrations for each of the $C$ classes. Thus, the selection cost is $T_{input} \times H \times C$. Assume that the average tokens of task-level thinking steps is $T_{task}$. In each iteration of the refinement phase, the teacher agent needs to predict each hard demonstration with cost $T_{task} + 2T_{input}$ based on task-level thinking. There is a degree of simplification, but roughly the input tokens are all $T_{input} + T_{task}$ and the output tokens are all $T_{task}$. The student agent needs to update task-level thinking steps based on test input with cost $T_{task} + 2T_{input}$. There is a degree of simplification, but roughly the input tokens are all $T_{input} + T_{task}$ and the output tokens are all $T_{task}$. Then the refinement cost is $2R \times (T_{task} + 2T_{input}) \times C$, where $R$ is the number of iterations for each hard demonstration.

In the inference phase, the cost of zero-shot prompt (Default) is $T_{input}$, zero-shot prompt with "Let's think step by step" (Auto) is $T_{input} + T_{task}$, and prompt with task-level thinking steps (PRTS) is $T_{input} + 2T_{task}$.

The total cost with $N_{test}$ test inputs is summarised in the Table 6. Taking i2b2 dataset as an example, $H = 3$, $C = 8$, $R = 4$, and $T_{task}$ is approximately $2T_{input}$. For practical applications, $N_{test}$ is much larger than 168, so generation tokens can be ignored. The task-level thinking steps introduced in the inference phase may need to be improved in the future.

## 5 Related Work

### 5.1 In-context Learning

It is widely investigated that severe bias exists in the output of LLMs for different demonstrations (Lu et al., 2022a; Turpin et al., 2023). Zhao et al. (2021) proposed a calibration method that fits the calibration parameters for uniform content-free input "*N/A*" is uniform across choices. More studies try to select specific demonstrations based on different philosophies to reduce bias, such as similarity (Liu et al., 2021) selecting similar examples to facilitate analogical learning, high uncertainty to improve inference efficiency (Diao et al., 2023) selection. Min et al. (2022) showed that LLMs could largely ignore the mapping between inputs and labels, while Webson and Pavlick (2022) showed that LLMs perform well on the NLI dataset with correct demonstrations even if the instruction is irrelevant or misleading.

## 5.2 Chain-of-Thought

Recent works have shown that LLMs can output better responses for ICL with high-quality explanation (Lampinen et al., 2022). Manual chain-of-thought (Wei et al., 2022) could prompt LLMs to generate step-by-step solutions, which leads to substantial improvements on many reasoning-intensive tasks. However, the performance improvement depends on manual efforts. The zero-shot CoT (Kojima et al., 2022) enables LLMs to automatically generate explanations. For unlabelled demonstrations, auto-CoT (Zhang et al., 2023b) partitions demonstrations into a few clusters, selects a representative demonstration from each cluster, and generates explanations using zero-shot CoT. For labeled demonstrations, Shum et al. (2023) optimizes a set of latent variables to select the most helpful and suitable demonstrations with correct self-generated answers. Except for the first-try CoT, Zheng et al. (2023) used previous outputs as hints to progressively guide toward the correct answers. Madaan et al. (2023) provided multi-aspect feedback on the previous output and refined it into new output. Our framework is also iterative, but it works on the task level and is based on feedback from hard demonstrations.

## 5.3 Prompt Engineering

Prompt offers a natural and intuitive interface for humans to interact with language models. For parameter-accessible models, gradient-based methods are widely adopted (Shin et al., 2020; Qin and Eisner, 2021; Lester et al., 2021). In the era of LLMs, searching directly in the natural language hypothesis is more popular. Honovich et al. (2022) found LLMs can generate the task instruction with several demonstrations. Zhou et al. (2022) automatically generates many instructions for the given demonstrations and selects the one with the maximum score function. The concept of task-level thinking steps is also part of prompt engineering, from simple "*Let's think step by step*" (Kojima et al., 2022) to well-designed plan-and-solve prompt (Wang et al., 2023).

## 6 Conclusion

Although LLMs have shown incredible performance on many tasks, we pointed out that many classification tasks, such as multifaceted analysis of subjective text, fine-grained classification, and domain-specific classification tasks are still challenging. Therefore, we proposed the concept of task-level thinking steps and verified the robustness for biased demonstrations. To further enhance the classification capabilities of LLMs, we designed a progressive revision framework, which purposefully improves the thinking steps based on feedback from hard demonstrations. Experimental results proved the superiority of our proposed method. Besides, with task-level thinking steps, we found that the automatically generated CoTs are more reasonable and effective.

## Limitations

**Task coverage.** We have proposed three challenging classification tasks: multifaceted analysis of subjective text, fine-grained text classification, and medical relation classification. Although there are many nontrivial findings, they are limited to traditional classification tasks. Future work would extend to a wider range of classification tasks, such as natural language inference and multiple choice.

**Experimental analysis and explanation.** Although many ablation experiments, robustness experiments and case studies have been conducted, there are still some aspects that have not been discussed. As an example, we adopt a default revising order for each task. While the checking mechanism reduces the negative impact of revision, there may still be other unforeseen challenges.

**Reliability.** Although our progressive revision framework has been validated on several challenging datasets, LLMs remain sensitive to small changes in the thinking steps. Future work would improve the reliability of the thinking steps by refining hard demonstrations selection, thinking steps revision and checking mechanisms.

**Unlabeled training sets.** The absence of massively labeled training sets is indeed a practical problem. However, we think there are still feasible solutions to obtain hard demonstrations. For example, measure uncertainty by running zero-shot prompts multiple times. If LLMs have the same outputs multiple times for the same train input, it means that the LLMs have high confidence. Either completely correct or completely wrong due to bias. On the contrary, if LLMs have very different outputs for the same input, it means that LLMs hesitate between certain classes and this train input

can be considered a hard demonstration. Annotate them and continue to use the proposed method in this paper.

## Ethics Statement

## Acknowledgements

This work was supported by the National Key Research and Development Program of China (2018YFC0830400) and the Shanghai Science and Technology Innovation Action Plan (20511102600).

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

## A  Details of Biased Demonstrations

Here are three biased demonstrations omitted in Table 1:

For domain abortion, Our #TruthTour cont' in the @user as we show the humanity of the unborn & inhumanity of abortion!

For domain feminist, Feel things will truly be equal in marriage when I see jock strap tossed to the 'single women' a weddings alongside the garter

For domain hillary, user Here's to fearless women chasing their goals.

## B  Details of Classification Tasks

We present the statistics of challenging classification tasks in Table 7, description in Table 8, prompt template in Table 9.

| Task | Dataset | Train | Dev | Test | Sampled Test | Labels |
|---|---|---|---|---|---|---|
| Multifaceted analysis of subjective text | Stance16 | 2620 | 294 | 1249 | 500 | 3 |
| | Emotion20 | 3257 | 374 | 1421 | 500 | 4 |
| Topic classification | 20news | 10.2k | 1.1k | 7.53k | 500 | 20 |
| Medical relation classification | i2b2 | 2808 | 312 | 6193 | 500 | 8 |
| | ChemProt | 3370 | 2030 | 2910 | 500 | 5 |
| | DDI | 3598 | 400 | 976 | 500 | 4 |

Table 7: Statistics of the number of samples and labels of challenging datasets.

| Dataset | Description |
|---|---|
| Stance16 | Please perform stance detection task.
**classes**: (A) none (B) against (C) favor
**PRTS**: 1. Read and understand the context: Carefully read the given statement or context to fully comprehend its meaning and the topic it addresses.2. Identify the domain: Determine the domain or subject matter the context is related to, such as politics, environment, social issues, etc. Make sure the domain matches the one specified in the question or task. 3. Determine the stance: Consider the context's position in relation to the domain and its implications, not just the general topic. Look for keywords, phrases, or sentiments that indicate a position in favor, against, or neutral. 4. Consider the broader implications: Assess how the context's position on the specific issue relates to the broader domain. For example, if the context supports a woman's right to choose in the domain of abortion, this would imply a stance in favor of abortion rights. 5. Choose the most appropriate stance: Based on your analysis, select the stance that best represents the position of the context in relation to the specified domain from the given choices (A) none, (B) against, or (C) favor. If the context is not relevant to the domain, choose (A) none. |
| Emotion20 | Please perform emotion recognition task.
**classes**: (A) anger (B) joy (C) sadness (D) optimism
**PRTS**: 1. Read and understand the given context. 2. Identify the emotions expressed in the context. Pay attention to that (B) Joy is a transient feeling of happiness and fulfillment (C) while optimism is a lasting positive outlook on the future. If typically associated with a low mood, a lack of energy, and a tendency to withdraw from social interactions. Consider (C) sadness rather than (D) 3. Match the identified emotions with the given choices. 4. Select the most appropriate choice as the answer. |

| Dataset | Description |
|---|---|
| 20news | Please perform topic classification task from choices.
**manual classes**:
(A) alt.atheism includes discussions and articles related to atheism, atheistic beliefs, and criticisms of religion (B) comp.graphics covers topics related to computer graphics, including discussions on image processing, rendering, and computer-aided design (CAD) (C) comp.os.ms-windows.misc focuses on discussions related to the Microsoft Windows operating system, including troubleshooting, software recommendations, and general Windows-related topics (D) comp.sys.ibm.pc.hardware revolves around discussions related to IBM-compatible PC hardware, including topics like motherboards, processors, memory, and peripherals (E) comp.sys.mac.hardware deals with discussions related to Apple Macintosh hardware, including Mac models, peripherals, and troubleshooting (F) comp.windows.x focuses on the X Window System, a widely used Unix-like operating systems, including System configuration, applications, and programming (G) misc.forsale includes postings of items for sale, ranging from electronics to household goods (H) rec.autos covers discussions related to automobiles, including car models, maintenance, buying/selling advice, and automotive technologies (I) rec.motorcycles includes discussions related to motorcycles, covering topics such as different motorcycle models, maintenance, safety, and riding experiences (J) rec.sport.baseball focuses on discussions related to baseball, including game analysis, player statistics, team performance, and baseball news (K) rec.sport.hockey covers discussions related to ice hockey, including game analysis, player statistics, team performance, and hockey news (L) sci.crypt revolves around discussions related to cryptography, encryption algorithms, and cryptographic protocols (M) sci.electronics deals with discussions related to electronics, including electronic circuits, components, and troubleshooting (N) sci.med includes discussions related to medical topics, including diseases, treatments, healthcare practices, and medical research (O) sci.space focuses on discussions related to space exploration, astronomy, and topics related to outer space (P) soc.religion.christian covers discussions related to Christianity, including theological debates, biblical interpretations, and religious practices (Q) talk.politics.guns revolves around discussions related to gun ownership, gun control policies, and the Second Amendment in the United State (R) talk.politics.mideast includes discussions related to politics in the Middle East, covering topics such as conflicts, peace negotiations, and regional dynamics (S) talk.politics.misc covers discussions related to politics that do not fit into the other specific political categories (T) talk.religion.misc includes discussions related to religion that do not fit into the other specific religious categories
**default classes**: (A) alt.atheism (B) comp.graphics (C) comp.os.ms-windows.misc (D) comp.sys.ibm.pc.hardware (E) comp.sys.mac.hardware (F) comp.windows.x (G) misc.forsale (H) rec.autos (I) rec.motorcycles (J) rec.sport.baseball (K) rec.sport.hockey (L) sci.crypt (M) sci.electronics (N) sci.med (O) sci.space (P) soc.religion.christian (Q)talk.politics.guns (R) talk.politics.mideast (S)talk.politics.misc (T) talk.religion.misc |

| Dataset | Description |
|---|---|
| i2b2 | What's the relation between <e1>treatment or medical test or medical problem</e1> and <e2>medical problem</e2>?

**classes**:
(A) <e1>Treatment</e1> improves <e2>medical problem</e2> (B) <e1>Treatment</e1> worsens <e2>medical problem</e2> (C) <e1>Treatment</e1> causes <e2>medical problem</e2> (D) <e1>Treatment</e1> is administered for <e2>medical problem</e2> (E) <e1>Treatment</e1> is not administered because of <e2>medical problem</e2> (F) <e1>Medical Test</e1> reveals <e2>medical problem</e2> (G) <e1>Medical Test</e1> conducted to investigate <e2>medical problem</e2> (H) <e1>Medical problem</e1> indicates <e2>medical problem</e2>

**PRTS**: 1. Read the context carefully and identify the e1 and e2 elements. 2. Understand the meaning of e1 and e2 in the context. 3. Analyze the relationship between e1 and e2 based on the context. Keep in mind that the relationship could be between medical problems, treatments, or tests. Pay close attention to whether the relationship implies improvement, worsening, causation, or a decision not to administer a treatment or test. Also, consider if the treatment is ineffective or has an unintended effect on the medical problem. 4. Compare the relationship with the given choices and select the one that best matches the context. If the relationship is between two medical problems, consider choice (H) as a possible answer. If the context suggests a decision not to administer a treatment or test, consider choice (E) as a possible answer. If the treatment is ineffective or worsens the medical problem, consider choice (B) as a possible answer. Be cautious not to confuse choice (B) with choice (D) when the treatment is administered but does not have the desired effect on the medical problem. |

| Dataset | Description |
|---|---|
| ChemProt | What's the relation between <e1>Chemical</e1> and <e2>Protein</e2>? 
 **classes**: 
 <e1>Chemical</e1> upregulation, activate, promote, increase activity of <e2>Protein</e2> 
 <e1>Chemical</e1> downregulation, inhibitor, block, decrease activity of <e2>Protein</e2> 
 <e1>Chemical</e1> is agonist of <e2>Protein</e2> 
 <e1>Chemical</e1> is antagonist of <e2>Protein</e2> 
 <e1>Chemical</e1> is the substrate metabolic of <e2>Protein</e2> 
 **PRTS**: 1. Read the context and identify the key terms (e1 and e2). 2. Understand the relationship between e1 and e2 based on the context. If the relationship is not explicitly stated, try to infer it from the information provided, including any indirect indications of their interaction. 3. Carefully analyze the inferred relationship and ensure it aligns with the context. Pay attention to the specific actions or effects of the chemical on the protein, or any other relevant information. 4. Match the relationship or inferred relationship with one of the given choices (A, B, C, D, or E), considering the specific actions or effects mentioned in the context and any indirect indications of their interaction. 5. Select the appropriate choice as the answer, considering both explicit and inferred information from the context, and double-check to ensure it accurately reflects the relationship between e1 and e2. |
| DDI | What's the relation between <e1>Drug</e1> and <e2>Drug</e2>? 
 **manual classes**: 
 Pharmacokinetics mechanism between <e1>Drug</e1> and <e2>Drug</e2>, the process of absorption, distribution, metabolism and excretion of a drug. 
 Pharmacodynamics mechanism between <e1>Drug</e1> and <e2>Drug</e2>, i.e., the mechanism of action and effect of the drug in the body. 
 A recommendation or advice regarding a interaction between <e1>Drug</e1> and <e2>Drug</e2> is given. 
 An interaction between <e1>Drug</e1> and <e2>Drug</e2> appears without providing any additional information. 
 **default classes**: Pharmacokinetics, Pharmacodynamics, advise, int 
 **PRTS**: 1. Carefully read the context sentence and identify the relationship between the two drugs mentioned. Pay close attention to the specific details of the interaction, such as metabolism, mechanism of action, recommendations, or lack of additional information. 2. Review the given choices and understand the differences between them. 3. Compare the relationship described in the context sentence with the given choices, focusing on the specific details of the interaction or the absence of such details. If there is an recommendation or advice regarding an interaction between the drugs, consider (C). If the emphasis is on the action of the drug in the body, choose (B) Pharmacodynamics: mechanism of action and effect of the drug in the body. (D) Without any additional information, consider (D). 4. Select the appropriate choice based on the context and the definition provided in the choices, ensuring that it aligns with the details mentioned in the context sentence or the lack thereof. Double-check your selection before finalizing your answer, and consider whether the chosen option truly reflects the information provided in the context. |

Table 8: Description of tasks.

| Setting | Default Template | Auto/Random/Similar Templates | Other Templates |
|---|---|---|---|
| zero-shot | {question}
"classes": {classes}
"input": {input}
"answer": | {question}
"classes": {classes}
"input": {input}
Let's think step by step. | {question}
"classes": {classes}
"input": {input}
"thinking steps": {thinking steps}
Let's think step by step. |
| few-shot | {question}
"classes": {classes}
"input": {input}
"answer": {answer}
...
"input": {input}
"answer": | {question}
"classes": {classes}
"input": {input}
"answer": {answer}
...
"input": {input}
Let's think step by step. | {question}
"classes": {classes}
"input": {input}
"answer": {answer}
...
"input": {input}
"thinking steps": {thinking steps}
Let's think step by step. |
| few-shot-CoT | / | {question}
"classes": {classes}
"input": {input}
"answer": {answer with CoT}
...
"input": {input}
Let's think step by step. | {question}
"classes": {classes}
"input": {input}
"answer": {answer with CoT}
...
"input": {input}
"thinking steps": {thinking steps}
Let's think step by step. |

Table 9: Prompt templates. Elements in braces {} are replaced with question-specific values.