# OpenReview forum: "Task-Level Thinking Steps Help Large Language Models for Challenging Classification Task"
_EMNLP/2023/Conference — EMNLP 2023 Main_

### Official Review · Reviewer_bGCR · 2023-08-01

**Soundness:** 3

**Excitement:**

4: Strong: This paper deepens the understanding of some phenomenon or lowers the barriers to an existing research direction.

**Missing References:**

Pryzant et al. Automatic Prompt Optimization with "Gradient Descent" and Beam Search (not required to be cited)

**Paper Topic And Main Contributions:**

This paper tries to solve classification tasks by creating an automatic chain-of-thought (COT) that is task-specific, instead of the general let’s think step by step COT or the data-specific COT that human labels. They named this as task-level thinking steps which reduce the bias introduced by demonstration and help LLMs to clarify tasks. The author also designs a progressive revision framework, which uses a teacher LLM and student LLM to generate and improve the thinking steps based on feedback from hard demonstrations. To demonstrate effectiveness, the author evaluates the proposed method on three kinds of challenging classification tasks: multifaceted analysis of subjective text, fine-grained similar-class text classification, and medical domain-specific classification. They showed that the proposed Progressively Revised Thinking Steps (PRTS) method outperforms competitive baselines in zero-shot, few-shot, and few-shot-CoT settings.

**Questions For The Authors:**

A Given that the order of demonstration could change the LLM prediction when in context learning, could the order of Hard Demonstration in figure 2 affect the final prediction in your method?

B As an analogy to traditional backdrop finetuning, we could consider task-level thinking steps as the parameter in the student model. How do you know when to stop training and loss coverage in your method?

**Reasons To Accept:**

1 The proposed method outperforms let’s think step by step COT and other baselines in all classification proposed.

2 Table 3 shows that PRTS reduced the bias introduced by demonstration significantly.

3 Even though the model is only “trained” on hard demonstrations, however, it does not “overfit” to the corrected classes in hard demonstrations and still knows how to assign to other calsses. This is shown in Figure 4.



**Reasons To Reject:**

1 Missing comparison with Madaan etal. (2023) to demonstrate the effectiveness of hard demonstrations. Does hard demonstrations method outpeforms all demonstrations method?

2 The task-level thinking steps are updated data by data during PRTS by the teacher. How do you ensure the tokens in task-level thinking steps generated from previous data would not be overwritten by the next data?

3 Most results only ran once, please include standard deviation.

4 Missing code to reproduce, and also lack details to differ from previous work. For example, is 1st call in line197 and 2nd call in line 200 implemented as two separate question answer calls to GPT (like step1-feedback and step2-refine in self-refine), or chat call where the 2nd call has access to both [human message and AI message](https://python.langchain.com/docs/modules/model_io/models/chat/) of the 1st call


**Reproducibility:**

2: Would be hard pressed to reproduce the results. The contribution depends on data that are simply not available outside the author's institution or consortium; not enough details are provided.

**Reviewer Confidence:**

3: Pretty sure, but there's a chance I missed something. Although I have a good feel for this area in general, I did not carefully check the paper's details, e.g., the math, experimental design, or novelty.

---

> ### Author Rebuttal · Authors · 2023-08-28
>
> We sincerely thank the reviewer and address all concerns below.
>
> **1. Missing comparison with Madaan etal. (2023) to demonstrate the effectiveness of hard demonstrations. Does hard demonstrations method outpeforms all demonstrations method?**
>
> Self-refine (Madaan et al., 2023) needs to manually design task-specific feedback and refine each test input of the task. Our approach uses simple feedback “*Please revise the thinking steps to clarify.*” and select hard demonstrations to obtain task-level thinking steps. No iterative refinement is required for each test input. Besides, self-refine is better suited for generative tasks (dialogue response generation, acronym generation, sentiment reversal), whereas our approach is better suited for classification tasks.
>
> **2. The task-level thinking steps are updated data by data during PRTS by the teacher. How do you ensure the tokens in task-level thinking steps generated from previous data would not be overwritten by the next data? Given that the order of demonstration could change the LLM prediction when in context learning, could the order of Hard Demonstration in figure 2 affect the final prediction in your method?**
>
> The order of hard demonstrations does matter for each round of updates. For example, the generated prompt "*If unsure, choose choice (B)*" may significantly reduce the accuracy of the other classes. However, such overfitted prompts are usually corrected by later rounds. Alternatively, we employ a checking mechanism that selects the prompt with the least negative impact on the rest of the samples (classes) from among the $K$ candidates. This is discussed from line222 to line232 and Figure 4.
>
> **3. Most results only ran once, please include standard deviation.**
>
> We have added standard deviation in 3 runs for the zero-shot setting below.
>
> | |Stance16  | Emotion20 | 20news | i2b2 | ChemProt | DDI
> |:---:|:---:|:---:|:---:|:---:|:---:|:---:|
> |Auto | $47.62\pm4.01$  | $72.81\pm2.08$ | $39.97\pm0.72$ | $29.63\pm6.59$ | $54.47\pm6.35$ | $36.22\pm5.51$ |
> |PRTS|$52.38\pm6.39$|$75.76\pm5.75$|$67.20\pm7.28$|$54.99\pm4.36$|$68.33\pm2.36$|$57.36\pm3.17$|
>
> **4. Missing code to reproduce, and also lack details to differ from previous work. For example, is 1st call in line197 and 2nd call in line 200 implemented as two separate question answer calls to GPT (like step1-feedback and step2-refine in self-refine), or chat call where the 2nd call has access to both human message and AI message of the 1st call**
>
> The 2nd call accesses to both human message and AI message of the 1st call. We found that the two-stage process of analyzing the student agent's output and then correcting it was more efficient and robust than the single-stage prompt "*Please analyze the student's output and revise the reflection steps to clarify*". This is different from the self-refine (Madaan et al., 2023), which iterates several times until the score exceeds the threshold. Our approach does not iterate on this revision. If the prediction is correct, the revision is accepted; otherwise re-analyzed.
>
> **5. As an analogy to traditional backdrop finetuning, we could consider task-level thinking steps as the parameter in the student model. How do you know when to stop training and loss coverage in your method?**
>
> For the $i$-th update, we use a candidate set of size 3 and choose the one that performs best on $\\{x_0, ..., x_{i-1}\\}$. This is similar to the greedy selection of the direction with the largest gradient in SGD. Adopting this checking mechanism makes the convergence smoother, as shown in Fig. 4. In our experiments, correcting one hard demonstration of each class provides sufficient improvement.

---

### Official Review · Reviewer_ZJta · 2023-08-02

**Typos Grammar Style And Presentation Improvements:** 1. The authors can elaborate more abo…
**Soundness:** 4

**Excitement:**

3: Ambivalent: It has merits (e.g., it reports state-of-the-art results, the idea is nice), but there are key weaknesses (e.g., it describes incremental work), and it can significantly benefit from another round of revision. However, I won't object to accepting it if my co-reviewers champion it.

**Missing References:**

1. Wang X, Wei J, Schuurmans D, et al. Self-consistency improves chain of thought reasoning in language models[J]. arXiv preprint arXiv:2203.11171, 2022.
2. Saunders W, Yeh C, Wu J, et al. Self-critiquing models for assisting human evaluators[J]. arXiv preprint arXiv:2206.05802, 2022.
3. Yao S, Zhao J, Yu D, et al. React: Synergizing reasoning and acting in language models[J]. arXiv preprint arXiv:2210.03629, 2022.


**Paper Topic And Main Contributions:**

This paper proposes a new form of Chain-of-Thought (CoT) prompt design by using GPT-4 to improve the thinking iteratively steps for the demonstration data. Compared with naive zero-shot CoT (i.e., let's think step by step), this method leverages the data labels as a signal to progressively revise the CoT texts so as to cover more nuanced explanations for the hard & biased demonstration data samples. Given the new CoT prompts, the paper shows that text-davinci003, gpt3.5, gpt-4 can all be significantly improved than the baseline on the 7 different classification tasks, such as stance detection and emotion recognition.

**Questions For The Authors:**

For the "conversations" between teacher and student, did the authors put the whole dialog history as part of the prompts for hard demonstrations, or only include the last revised turn of the task-level thinking steps?

**Reasons To Accept:**

1. Propose a set of new prompt templates for text classification tasks. The progressive revision framework uses the data labels as supervision to continually revise and improve the chain-of-thought prompts in an automatic manner. With those generated detailed demonstrations as input, the LLMs can be more robust to biased data samples

2. This work also ensembles a wide range of downstream tasks, including multifaceted analysis of subjective text, fine-grained topic prediction, and vertical domain relation classification, which spans various scenarios. This benchmark could potentially serve as a standard testbed for the evaluation of LLMs or any prompt engineering methods.

**Reasons To Reject:**

1. Unlike the simple zero-shot CoT, the new CoT prompt method may not be easy to transfer to more tasks. Since it requires manual design for the prompt words, like "identify the two medical terms" , "identify the domain xxx" and "determine the stance of statement", those template hinders more general-purposed text classification tasks.

2. Lacking stronger baselines, such as ReAct, Reflex, self-critque, self-refine, and AutoGPT. Those are more general design prompt engineering techniques and also can take the demonstration labels as supervised signals to continually revise and review LLM's output, thus are able to generate richer CoT demonstrations for in-context learning. Although I understand some of the baselines are contemporary work, only comparing PRTS with "let's think step by step" is too weak.







**Reproducibility:**

4: Could mostly reproduce the results, but there may be some variation because of sample variance or minor variations in their interpretation of the protocol or method.

**Reviewer Confidence:**

3: Pretty sure, but there's a chance I missed something. Although I have a good feel for this area in general, I did not carefully check the paper's details, e.g., the math, experimental design, or novelty.

---

> ### Author Rebuttal · Authors · 2023-08-28
>
> We sincerely thank the reviewer and address all concerns below.
>
> **1. Unlike the simple zero-shot CoT, the new CoT prompt method may not be easy to transfer to more tasks. Since it requires manual design for the prompt words, like "identify the two medical terms" , "identify the domain xxx" and "determine the stance of statement", those template hinders more general-purposed text classification tasks.**
>
> Thanks for your comments. In general, "determine the stance of statement" etc. are task goals or descriptions, which are usually known a priori [1]. The difficulty lies in what prompts can align the knowledge/bias of  LLMs (introduced by pre-training and RLHF) [2] with task goals. In this paper, we argue that prompts that elucidate the differences between categories learned from hard demonstrations are particularly important for classification tasks.
>
> **2. Lacking stronger baselines, such as ReAct, Reflex, self-critque, self-refine, and AutoGPT. Those are more general design prompt engineering techniques and also can take the demonstration labels as supervised signals to continually revise and review LLM's output, thus are able to generate richer CoT demonstrations for in-context learning. Although I understand some of the baselines are contemporary work, only comparing PRTS with "let's think step by step" is too weak.**
>
> Thank you very much for the provided relevant work! This issue has been partially discussed in Question2 of the last review. We tried to reproduce the above methods. AutoGPT is a great approach in the unlabelled few-shot setting. **Self-critque** [3] allows LLMs to assist in evaluating complex tasks by fine-tuning simple tasks that humans can evaluate. This is similar to our approach to gaining a deeper understanding of tasks by allowing LLMs to distinguish between choices. However, our approach belongs to prompt engineering and does not require fine-tuning. **Auto** in few-shot-CoT in Table 2 is actually a labeled version of **AutoGPT** [4] but is still inferior to our method. We will complete the related work and provide as many baseline comparisons as possible.
>
> **3. For the "conversations" between teacher and student, did the authors put the whole dialog history as part of the prompts for hard demonstrations, or only include the last revised turn of the task-level thinking steps?**
>
> Considering the computational cost, historical dialogues are not included in the prompt. Also, the initial prompt is always on while the task-level thinking steps are constantly updated.
>
> **4. The authors can elaborate more about why choosing multifaceted subjective text analysis etc, as challenging classification tasks. There are also many other difficult classification tasks like BigBench, MathQA, MSQA, HotPotQA, etc.**
>
> Thanks for your advice. The tasks mentioned above are "multiple choice" whereas this paper focuses on "classification". The difference is that the choices are always the same for all the examples for the classification task. Therefore, our task-level thinking steps can explicitly clarify the difference between the choices and prompt all examples.
>
> **5. Table 1 & 4, what does the different color mean? May need to explain in the captions.**
>
> Thanks for your kind advice. Red text indicates incorrect or misleading keywords, while blue text indicates correct or enlightening keywords in the prompt. Additional captions will be added to the revised manuscript.
>
> [1] Zhang W, Deng Yue, Liu B, et al. Sentiment Analysis in the Era of Large Language Models: A Reality Check. arXiv preprint arXiv:2305.15005, 2023
>
> [2] Si C, Friedman D, Joshi J, et al. Measuring Inductive Biases of In-Context Learning with Underspecified Demonstrations. ACL, 2023
>
> [3] Saunders, W, Yeh, C, Wu, J, et al. Self-critiquing models for assisting human evaluators. arXiv preprint arXiv:2206.05802.
>
> [4] Zhang, Z, Zhang, A, Li, M, et al. Automatic chain of thought prompting in large language models. arXiv preprint arXiv:2210.03493.

---

### Official Review · Reviewer_J9Vu · 2023-08-18

**Typos Grammar Style And Presentation Improvements:** N/A
**Soundness:** 4

**Excitement:**

3: Ambivalent: It has merits (e.g., it reports state-of-the-art results, the idea is nice), but there are key weaknesses (e.g., it describes incremental work), and it can significantly benefit from another round of revision. However, I won't object to accepting it if my co-reviewers champion it.

**Missing References:**

N/A

**Paper Topic And Main Contributions:**

This paper proposes using task-level thinking steps and a progressive revision framework to improve the performance of large language models (LLMs) on challenging classification tasks. The key ideas are:
* Introducing task-level thinking steps to guide the LLM's reasoning process and reduce bias from demonstrations.
* Progressively revising the thinking steps by having one LLM agent correct another on hard examples, in order to enhance the distinction between confusing classes.

Main contributions:
1. Proposes the concept of task-level thinking steps to help debias LLM predictions and clarify the reasoning process.
2. Designs a framework where one LLM agent revises the thinking steps based on feedback from another LLM agent on hard examples.
3. Shows improved performance over baselines in zero-shot, few-shot, and few-shot-CoT settings on classification tasks requiring multifaceted text analysis, fine-grained distinction, and domain knowledge.
4. Demonstrates the framework's robustness to biased demonstrations and ability to generate better chain-of-thought explanations.

**Questions For The Authors:**

Refer to weakness.

**Reasons To Accept:**

1. Introduces a intuitive technique of task-level thinking steps tailored to classification tasks, advancing prompt engineering for LLMs.
2. Progressive revision framework purposefully enhances LLM capabilities based on feedback, unlike standard pre-training or fine-tuning.
4. Ablation studies, robustness tests, and case analyses give useful insights into the framework's capabilities.
5. Allows generating higher quality chain-of-thought explanations without human involvement.

**Reasons To Reject:**

1. My concern is the cost issue. This iterative framework introduces extremely high token costs, the number of times and tokens to avoid inviting GPT-4 is much more compared to doing ICL directly with GPT-4s, and the ratio of boosts to token growth determines the potential of this approach in practice. And, I also care about the average number of iterations needed.
2. From a high-level perspective, the method and reflection [1] are strongly related and some discussion is suggested for the authors.
3. This approach requires traversing the training set (147l) to find difficult demonstration, and in the face of a small training set, it may be difficult to find challenging demonstration to generate task-level thinking.

[1] Reflexion: Language Agents with Verbal Reinforcement Learning. 2023.03.

**Reproducibility:**

4: Could mostly reproduce the results, but there may be some variation because of sample variance or minor variations in their interpretation of the protocol or method.

**Reviewer Confidence:**

4: Quite sure. I tried to check the important points carefully. It's unlikely, though conceivable, that I missed something that should affect my ratings.

---

> ### Author Rebuttal · Authors · 2023-08-28
>
> We sincerely thank the reviewer and address all concerns below.
>
> **1. My concern is the cost issue. This iterative framework introduces extremely high token costs, the number of times and tokens to avoid inviting GPT-4 is much more compared to doing ICL directly with GPT-4s, and the ratio of boosts to token growth determines the potential of this approach in practice. And, I also care about the average number of iterations needed.**
>
> Thanks for your careful observation, but the cost is inevitable for almost all methods. RL-based methods such as **ReAct** [1] and **Reflexion** [2] require the most cost because of trajectory memory and multiple explorations. **Self-Refine** [3] requires refining each input based on manual feedback. Our method only requires refining one hard demonstration per class on most datasets. Therefore, as many iterations as the number of classes is sufficient to get a satisfying task-level thinking step. No iterative refinement is required for each test input.
>
> **2. From a high-level perspective, the method and reflection are strongly related and some discussion is suggested for the authors.**
>
> Thanks for the enlightening insights. Feedback has been the consensus for improving the capabilities of LLMs. However, there are still differences between these approaches. **React** [1] and **Reflexion** [2] excel at multi-step tasks such as sequential decision-making and multi-step reasoning. The key is that the environment and feedback change at each step, fitting well into the reinforcement learning framework. **Self-refine** [3] is good at generative tasks such as dialogue response generation, acronym generation, and sentiment reversal. The key is the richness and plausibility of manually designed feedback. We argue that our contribution is focusing on classification tasks that generative LLMs are not very good at and trying to get LLMs to distinguish between classes through feedback.
>
> **3. This approach requires traversing the training set (147l) to find difficult demonstration, and in the face of a small training set, it may be difficult to find challenging demonstration to generate task-level thinking.**
>
> Thanks for your comments. LLMs have achieved performance close to supervised training for simple or publicly exposed tasks. For example, 0.936 accuracy for zero-shot ChatGPT while 0.932 accuracy for fine-tuned T5-large [4]. In this case, hard demonstrations are not easy to find, but applying our method is also unnecessary. We believe that our method is effective for domain-specific classification tasks because of the weaker performance of LLMs (only 0.5025 accuracy for Stance16 [4]).
>
> [1] Yao S, Zhao J, Yu D, et al. React: Synergizing reasoning and acting in language models[. arXiv preprint arXiv:2210.03629, 2022.
>
> [2] Shinn N, Cassano F, Labash B, et al. Reflexion: Language Agents with Verbal Reinforcement Learning. arXiv preprint arXiv:2303.11366, 2023.
>
> [3] Madaan, A, Tandon, N, Gupta, P, et al. Self-refine: Iterative refinement with self-feedback. arXiv preprint arXiv:2303.17651, 2023.
>
> [4] Zhang W, Deng Yue, Liu B, et al. Sentiment Analysis in the Era of Large Language Models: A Reality Check. arXiv preprint arXiv:2305.15005, 2023

---

### Meta-Review · Area_Chair_oqFH · 2023-09-25

**Recommendation:** 4

**Metareview:**

This paper introduces a useful and intuitive concept of task-level thinking steps to help LLMs overcome any bias brought in by demonstration examples in in-context learning setups. It also proposes a framework to revise the thinking steps for confusing classes in classification tasks, by having one LLM correct another on hard examples. It provides comprehensive experiments with several relevant baselines, and demonstrates robustness to biased demonstrations and ability to generate chain-of-thought explanations without human involvement.

Concerns include the generalisability of the prompt templates (and the manual effort in designing them) to other classification tasks, and the lack of sufficient baselines for more comprehensive comparisons.

For the authors — please include the cost analysis and a discussion on how to use the framework when the training sets are small in the paper. It would also be useful to open-source the code for future works on this effort.

---

### Decision · Program_Chairs · 2023-10-07

**Decision:**

Accept-Main

**Comment:**

This paper introduces a useful and intuitive concept of task-level thinking steps to help LLMs overcome any bias brought in by demonstration examples in in-context learning setups. It also proposes a framework to revise the thinking steps for confusing classes in classification tasks, by having one LLM correct another on hard examples. It provides comprehensive experiments with several relevant baselines, and demonstrates robustness to biased demonstrations and ability to generate chain-of-thought explanations without human involvement.

Concerns include the generalisability of the prompt templates (and the manual effort in designing them) to other classification tasks, and the lack of sufficient baselines for more comprehensive comparisons.

For the authors — please include the cost analysis and a discussion on how to use the framework when the training sets are small in the paper. It would also be useful to open-source the code for future works on this effort.